

# AndroAnalyzer: android malicious software detection based on deep learning

Recep Sinan Arslan

Department of Computer Engineering, Kayseri University, Kayseri, Türkiye

## ABSTRACT

**Background:** Technological developments have a significant effect on the development of smart devices. The use of smart devices has become widespread due to their extensive capabilities. The Android operating system is preferred in smart devices due to its open-source structure. This is the reason for its being the target of malware. The advancements in Android malware hiding and detection avoidance methods have overridden traditional malware detection methods.

**Methods:** In this study, a model employing AndroAnalyzer that uses static analysis and deep learning system is proposed. Tests were carried out with an original dataset consisting of 7,622 applications. Additional tests were conducted with machine learning techniques to compare it with the deep learning method using the obtained feature vector.

**Results:** Accuracy of 98.16% was achieved by presenting a better performance compared to traditional machine learning techniques. Values of recall, precision, and F-measure were 98.78, 99.24 and 98.90, respectively. The results showed that deep learning models using trace-based feature vectors outperform current cutting-edge technology approaches.

## INTRODUCTION

In recent years, smart devices have become the main medium of communication among people. While phones used to offer only verbal communication, they are now smart devices. This rich technological equipment enables users to make increasing use of these devices (*Feizollah et al., 2017*). In the past, phones were used to send SMS messages and make phone calls, but today they are used in many areas, primarily in web services and as a camera, music service provider, and tablet PC. These devices are equipped with hardware and various sensors with advanced memory and processing power, just like computers. For this reason, they are very easy to customize (*Yen & Sun, 2019*).

An operating system is required for users to use smart devices. At present, a variety, such as IOS, Android, Windows, and Blackberry OS, are available for smart devices (*Alzaylaee, Yerima & Sezer, 2020*). Android offers rich media support, optimized graphics infrastructure, and powerful browser support to its users. Along with working with different APIs, it also supports structures such as sensor usage and real-time map systems. The fact that the Android operating system offers such a wide range of free features that are open source has made it widely preferred among users (*Feizollah et al., 2017*).

Corresponding author
Recep Sinan Arslan,
sinanarslanemail@gmail.com

Applications that allow users to use more features can be developed and distributed via both Play Store and third-party environments (*Saif, El-Gokhy & Sallam, 2018*). Since these applications provide free benefits to users, Android has become the target of malware developers (*Feizollah et al., 2017*).

It is possible to repackage Android applications by adding different hidden and malicious codes to the development files in the binary source structure. Thus, users are likely to be exposed to these undesirable outcomes without even realizing it. For this reason, researchers are trying to develop different malware detection tools to detect these activities and to reveal applications prepared for malicious purposes.

In the Android operating system, if an application is desired to be used on a smart device, first the application is downloaded from the relevant platform, then the permissions required for the application to run are accepted, and finally the installation is performed. It is not possible to install and run an application without accepting all the permissions claimed to be required to install the application. Many malicious developers turn this situation to their advantage and request permissions such as camera access, access to text messages, and reading private information that the application does not need. Many Android users who do not have sufficient knowledge of computers accept these permissions and install the application unaware of this malicious purpose. Thus, they become the clear target of malicious activities (*Islam et al., 2020*).

Various techniques such as static and dynamic analysis and derivatives of these techniques have been proposed for the detection of malware and protection of end users (*Yen & Sun, 2019*). Information about static analysis, signature-based analysis, application, and expected behavior includes observations presented explicitly and implicitly in binary source code. It is a fast and effective method. However, there are other methods such as hiding the software code or detection bypass that developers use to circumvent this analysis method. Dynamic analysis, which is also known as behavior-based analysis, is the collection of information about the runtime of the application, such as system calls, network access requests, information changes in files, and memory installed in the real environment or on a sandbox (*Jerbi et al., 2020*).

Selecting meaningful features from Android applications using static analysis and modeling them in a better manner enable the development of a powerful malware tool. Thus, unlike dynamic analysis approaches, downloading of applications to devices is prevented even on a temporary basis for detection. The method suggested in the present study is new, with the purpose of learning the requested permissions required by the application with the proposed model and thus detecting the malicious activities of new applications. The recommended method can be used in applications produced for all Android versions, including Android 11 API 30. The following improvements have been made in the present study and the aim was to detect malware more accurately.

The contributions of the present work are as follows:

- Development of an advanced deep learning-based network for analyzing and developing malware for all Android versions.

- Preparation of specific malicious and benign application datasets for training of network structure.
- Via running of this process before installation, users are prevented from being exposed to any dangerous activities, even for a short time.
- Ensuring users are warned about these detected malwares and presenting a model that enables more perfect detection due to continuous learning and converges false positive (FP) and false negative (FN) values to the minimum.
- Demonstration of extracting combined features and using them in education is a better way than signature-based or behavior-based techniques.
- Demonstration of the precision and accuracy of the proposed approach in a comparison with different machine learning techniques.

The rest of this article is organized as follows. In "Literature Review", recent studies performed between 2018 and 2020 using static, dynamic, and machine learning techniques are mentioned. In "Materials and Methods", the methodological infrastructure of the model proposed in the present study, pre-processing processes, the Android application structure, and details of the original dataset prepared for use in the tests are given. The evaluation scheme, experimental parameters, and performance results of the proposed method are presented comparatively in "Results". In "Conclusions", the study is evaluated in general and recommendations for the future are given.

## LITERATURE REVIEW

In this section, recent studies related to Android malware detection, feature generation, and selection, static, dynamic, and machine learning approaches are discussed.

### Static analysis

The use of static analysis to determine whether Android applications are malicious or not is based on inspection of the application code and it remains popular. Using the static analysis approach, solutions were produced using permissions, API calls, command systems, and features based on the purpose of use. Although static analysis approaches allow more comprehensive code inspection, malware developers are able to use different techniques to avoid static analysis and to hide purposeful code. Data encryption, hiding, update attacks, or polymorphic techniques are examples of these hiding techniques (*Zhao et al., 2018*).

DAPASA is a graphical malware detection tool that calculates the sensitivity of API calls using a data mining technique called TF-IDF (*Fan et al., 2017*). Detection is performed based on two assumptions that indicate how sensitive API calls are called.

In *Shahriar, Islam & Clincy (2017)* a model was proposed to detect malicious software by analyzing the permissions requested in the mobile application. In the first stage, statistics of the permissions requested by most of the malware were produced and their usage intensities were determined. In the second stage, the application status of the permissions with high usage intensity was investigated in order to determine malicious behavior.

In *Arslan, Doğru & Barışçı (2019)* a malware detection tool with a code analysis base was developed to determine whether the permissions are requested by the application or not and whether these permissions are used or not. Classification was performed according to a statistically determined threshold value. A 92% success rate in Android malware detection was achieved.

In *Taheri et al. (2020)* a study was reported based on the extraction of properties including permission, purpose, and API calls of applications and classification of these extractions with K-nearest neighbors (KNN) derivative algorithms. It is a similarity-based approach. Malware was detected with an average accuracy of 90% with the proposed algorithm.

AppPerm analyzer (*Doğru & Önder, 2020*) is a malware detection tool based on examining the codes together with the manifest file, creating double and triple permission groups, and determining the risk scores of the applications accordingly. A TP value of 95.50% and a specificity value of 96.88% were achieved.

## Dynamic analysis

Dynamic analysis is another method used to detect security vulnerabilities in Android applications (*Zhao et al., 2018*). It involves a more complex process compared to the static analysis approach. Since the dynamic analysis approach is based on tracking the behavior of the application during runtime, it is not easy for malicious application developers to prevent this analysis approach. Researchers often use the dynamic analysis approach to overcome the problems they encounter during the static analysis approach. There are many studies that suggest a dynamic analysis model for Android malware detection. In this section, a number of current studies are mentioned.

MADAM is a malware detection tool that uses a signature- and behavior-based malware detection approach. It uses different properties to evaluate and classify applications. Features at four levels, i.e., application, kernel, user and package, were extracted and used in the study (*Saracino, Dini & Martinelli, 2018*).

In *Amamra et al. (2016)*, a malware detection mechanism using tracking behavioral system call traces was proposed. Malicious activities were predicted by examining the frequency of behavioral system calls with a previously trained classifier.

There are some points in which both static and dynamic analysis approaches are advantageous and disadvantageous. Static analysis is capable of finding malicious components and dynamic analysis is capable of observing the application's status at runtime. For this reason, some studies suggest both methods be used together in order to benefit from their advantages (*Surendran, Thomas & Emmanuel, 2020*; *Onwuzurike et al., 2018*).

## Machine learning

Machine learning is the approach of allowing algorithms to self-learn the necessary parameters from the data to predict malware detection in Android applications. Machine learning techniques (*Gibert, Mateu & Planes, 2020*), which are successfully applied in many problems today, have also been implemented in the field of mobile security with

deep learning in the present study. In this section, some of the studies that have been performed in recent years and that have used machine learning techniques are mentioned. Successful results were obtained in all of the studies.

AspectDroid (*Al-Gombe et al., 2018*) is a system that allows the monitoring of suspicious activities such as dynamic class loading during runtime and review of them afterwards by writing them to log files. Activity tracking code has been added to the source code of the applications for this process.

NTPDroid (*Arora & Peddoju, 2018*) is a model using Android permissions and network traffic-based features. The aim is to determine the probability of malicious behavior. It is possible to decrease the FP value and thus the level of evaluating benign practices as malicious practices.

In *Naeem et al. (2020)* a suggestion was made for the detection of malware on the Internet of Things. The in-depth analysis of malware is based on visualization by color image and classification with convolutional neural network (CNN) methodology. According to the experimental results, more successful results are produced compared to machine learning and old-style static and dynamic analysis approaches.

In *Arshad et al. (2018)* a hybrid malware detection model was created for Android devices. In this model structure, in the first step, features to be obtained by static analysis such as requested permissions, permissions used, hardware components, intent filters, and suspicious system calls were extracted. In addition, network usage and file read–write statistics were obtained by dynamic analysis of the application. Applications with these extracted features were classified by support vector machine (SVM) technique.

In *Xiao et al. (2019)* a deep learning-based detection method using system call sequences and created with an LSTM-based classifier was proposed. It yielded a 96.6% recall value with a 9.3% FP value.

In *Wang et al. (2018)* a model using permission patterns extracted from Android applications was proposed. Required and used permissions were used. While achieving the highest 94% classification performance, an FP value of 5% and FN value of 1% were obtained.

In *Varna Priya & Visalakshi (2020)* a model based on extraction of features based on static analysis by using manifest files of apk files and selection by KNN and classification by SVM algorithms of these features was proposed. With that method, a TP ratio of 70% and above and an FP value close to zero were obtained. Due to feature selection by KNN and classification by SVM, recognition performance close to that of classification models with deep learning was achieved.

In *Zhu et al. (2018)* sensitive API calls, permissions, permission usage rates, and properties obtained from system events were extracted. An ensemble rotation forest classification model was proposed and it yielded 88.26% accuracy, 88.40% sensitivity, and 88.16% precision values. With the proposed model, an improvement of 3.33% was achieved compared to the classification by SVM.

In *Farhan et al. (2019)* a CNN-based network model was proposed to test malware attacks on the Internet by visualizing color images. Improved performance results for cyber security threats were obtained in the models in which CNNs were used.

In *Hou et al. (2017)* the AutoDroid tool, which automatically detects malware according to API calls extracted using static analysis, was described. The system was developed using different types of deep neural networks (deep belief networks etc.). In the design made by DBN, a 95.98% success rate was achieved in the tests performed using 2500 benign and 2500 malicious applications.

Since Android is an open source and extensible platform, it allows us to extract and use as many application features as we want. The method proposed in the present study has a robust and scalable perception and uses a deep learning structure. In this manner, the method has successful detection ability together with low resource consumption. It is more successful than current deep learning-based Android malware detection tools. Moreover, it is based on real devices rather than emulators. Due to the 349 features extracted among the applications available in the dataset consisting of a total of 7,622 applications, more successful results were obtained compared to the existing models. It is a method that extensively investigates Android malware on real devices and comparatively evaluates different methods to measure the impact of codes.

# MATERIALS AND METHODS

## Methodology

In the studies we mentioned in the second section, it was shown that there are some problems related to both static analysis and dynamic analysis. In the model proposed in the present study, static analysis and machine learning techniques were used together for malware detection. In this manner, it became possible to achieve classification with a better success rate and to create a safer Android environment for users. All of this is carried out without the apk files being uploaded to the user's mobile devices.

A flow chart of the proposed model is shown in Fig. 1. In the first stage, the application datasets were created. Both malicious and benign datasets are need for training the model. In some of the applications used in the creation of these datasets, there may be problems in accessing the source code, and, in others, there may be problems in accessing the readable manifest.xml file. For this reason, first of all, these applications are determined and removed from the dataset before going on to the feature extraction stage. This process is applied for both malicious and benevolent applications.

After this preparation, application Java codes and application package files should be accessed in order to obtain the properties of applications with static analysis. In the second stage, these operations were performed by using Aapt, dex2jar, Xxd, jdcli.jar libraries. In this manner, access to the manifest file including the Java code files of the applications, permissions, intent filters, and manufacturer information was provided. Then a word-based parsing operation was performed on the manifest.xml file and feature vectors that would be used for training and that contained meaningful information about the application were obtained. At this point, separate programs/frameworks are used to access application codes and the manifest.xml file. The reason for this is that while the manifest file can be accessed with the aapt library, it is not possible to access Java codes with the same program. Similarly, Java codes can be only accessed with dex2jar, xxd, or jdcli.jar. The features to be used in the deep learning model were extracted by reading the data in

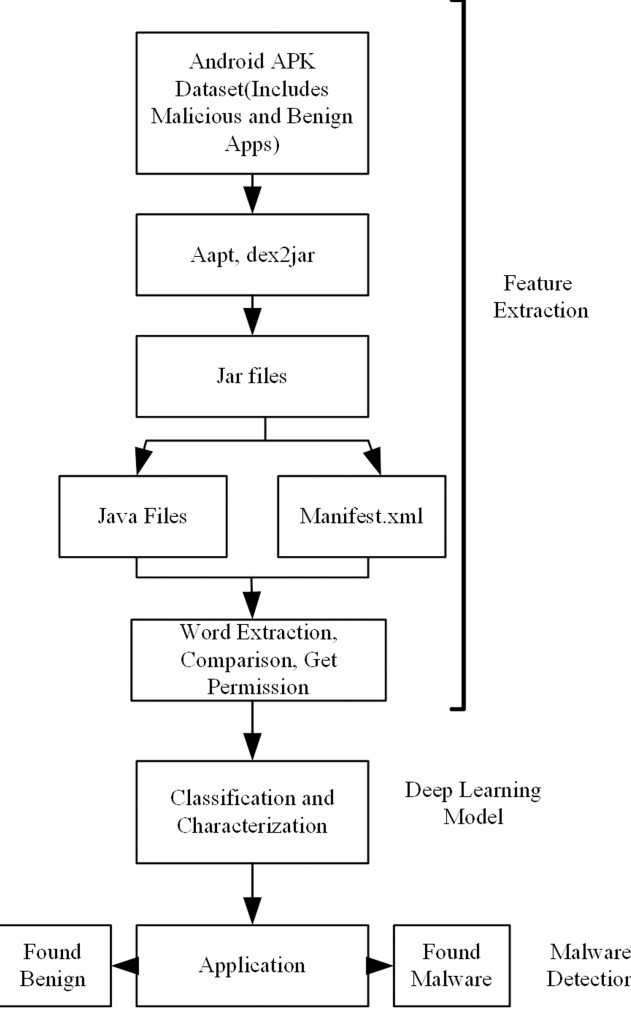

**Figure 1 Overview of the proposed model for android malware detection.**

both the code and the manifest file. A vector is produced from these features. Classification of test applications was made with the model obtained as a result of the training.

Details related to the success of the classification are compared in "Results".

## Android application basics

Android applications are mainly written in Java language and then they are compiled with data and source files into an archive file called an Android Application Kit (APK). The APK is the file that is distributed in the application market and used for installation of the application. There are four different types of application component: event, services, broadcast receiver, and content provider. Communication among these components is provided by using a messaging object called intent. In Android, applications must declare all of their components in an XML manifest file within the APK. Intent filters define the limits of each component's capabilities and they are included in the package. Additional information declared in the XML file includes the user permissions required by

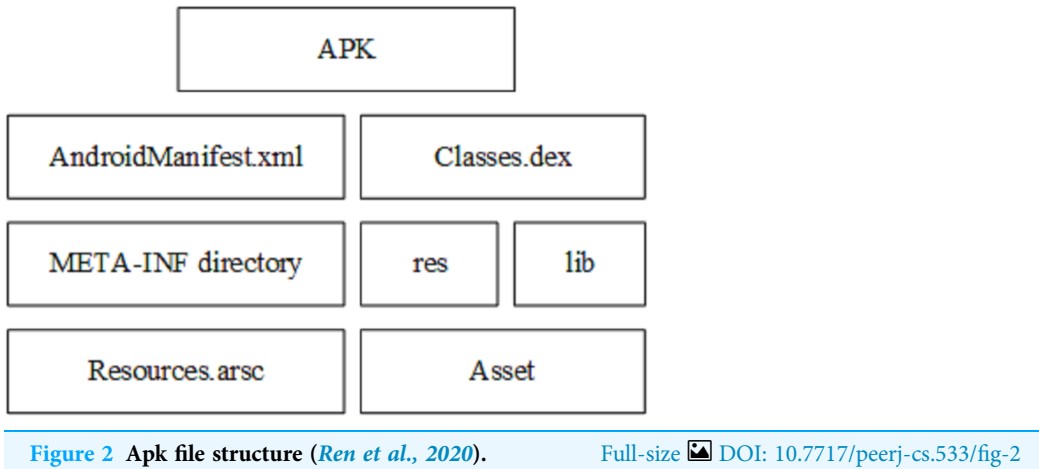

**Figure 2  Apk file structure (*Ren et al., 2020*).**     

the application (CALL_PHONE, SEND_SMS etc.), the minimum API level, and also the hardware and software features to be used by the application (GPS, camera, multitouch screen) (*Martin, Cabrera & Camacho, 2019*).

As shown in Fig. 2, in addition to the XML file, there are one or more dex files containing all classes in the APK file that will run on its own virtual machine (Dalvik Virtual Machine), the lib folder containing the files compiled specifically for the processor layer, application certificates, resource lists, and a META_INF folder that contains the SHA-1 summary of these resources and a resources.arsc file that contains the precompiled resources.

## Dataset description

One of the biggest challenges in Android malware detection research is that it is not easy to access an up-to-date and reliable dataset that is large enough. In the present study, the Drebin (*D Arp, 2012*) and Genome (*M Team, 2012*) malicious application datasets were used to obtain the malicious application set. The Drebin dataset contains 5560 malicious applications. It was created between August 2010 and October 2012. There are applications from 179 different application families (*Islam et al., 2020*). On the other hand, the benign application dataset was created specifically for this study. There are applications from different categories such as books/references, communication, finance, music/entertainment, news and magazines, media, sports, tools, transportations, and weather in the benign dataset. A homogeneous distribution was attempted.

Table 1 shows the details of the dataset used for the study. Applications labeled as unknown were not used. Out of the total 6,739 applications in the Drebin and Genome datasets, 6,661 were determined as malicious datasets and they were used in the training processes. For the benign dataset, 961 out of the total 1,073 applications were determined as truly benign and they were used in the present study.

Benign refers to applications identified after verifying that the applications are not malicious or do not have potentially unwanted functionality. To detect this situation, the website https://www.virustotal.com (*VT Team, 2020*), which contains more than one antivirus program and can perform scanning simultaneously through different programs,

**Table 1 Statistics of distribution of malware and benign applications in the datasets.**

| Dataset | | | |
|---|---|---|---|
| | Malicious | Unknown | Total |
| Drebin | 5,498 | 62 | 5,560 |
| Genome | 1,163 | 16 | 1,179 |
| | Benign | | |
| Proposed model original dataset | 961 | 112 | 1,073 |

was used. Creating this kind of benign label is a difficult and costly process. On the other hand, malicious applications are those that pose a potential danger to users. This application set was automatically tagged as it was taken from the Drebin and Genome datasets, which have been used in many studies before. Unknown is the name given to instances when malicious and benign tags are not assigned. These samples could not be processed and they could not be examined manually by accessing their codes. All existing examples were first placed in this category. After the examinations, they were transferred to the other two labels (benign, malicious). Most of these applications can be expected to be harmless, but some of them are likely to have malicious purposes as there are cases when they cannot be detected. For this reason, they were not directly included in either of the other two groups.

## Feature extraction and preprocessing

To develop an effective Android malware model, it is quite critical to obtain robust and broad representation features such as user permissions, manufacturer and user information, intent filters, process names, and binary files. With this information, it becomes possible to detect malware.

In the present study, the processes of obtaining source codes and binary files of applications by reverse engineering and converting application APK files to Java codes were carried out. The xxd tool was used to extract Dex files in APK files. Access to Java source codes of the applications was provided with the Dex2jar and jdcli.jar libraries. Moreover, the aapt dump AndroidManifest.xml tool offered by the Android SDK was used to obtain xml files. The keywords and permissions obtained from the AndroidManifest.xml file were used in the training of the deep learning model proposed in the present study. A total of 349 features were extracted and used in model training.

## RESULTS

In this section, the proposed model testing process is explained in detail. Both the obtaining of the best deep learning structure and comparison of it with other classification methods or similar studies are shown in the table.

## Experimental setup and parameters

In order to measure the performance and efficiency of the deep neural network model proposed, experiments with different parameters were conducted. A laptop with CORE I7

LZ5070, 8 GB RAM memory was used for training of the proposed model. Windows 10 64-bit operating system and an $\times$64-based processor were used to create the presented malware detection model. The training time of the proposed model varies depending on the complexity of the deep learning model and has 0.2 MB/s memory usage. In addition, Python with scikit-learn, pandas and NumPy packages were used for the experiments. The proposed model was evaluated with TensorFlow.

Many experiments were conducted to create the most successful model in classifier design for malware detection with a deep learning model. A distinction of 80%/20% was made, respectively, to use the model with 349 features in the training and testing stages. After this separation, under-sampling or over-sampling procedures were not used to balance the training data. In addition, although many permissions available in the Android OS are not used in most applications, no feature selection process was performed to ensure objectivity in future tests. After all, the original feature set and the original data vector were used in the training and testing phase without using any synthetic data generation for feature selection in the data. In order to obtain the most successful model in the DNN model, many different DNN models with different layers and nodes in each layer were created and tests were carried out. Thus, the best model was obtained. While the softmax function is used in the output layer, reLu is used as the activation function in the input and hidden layers. The Adam function is used for optimization in the output layer. Since the model completed the learning process in approximately 50 epochs, the training stage was terminated at this point. The best practice examples and the best values obtained as a result of numerous tests were used in the selection of parameters.

## Performance measure

The aim of the present study was to create a deep learning-based model for classifying malicious and benign applications and detecting malicious applications. In the experiments, our deep learning models were trained in a binary classification problem as benign or malicious. The created model included an original deep learning architecture. The effectiveness of the model was evaluated and demonstrated by creating a confusion matrix. As a result of these tests, performance values between different popular machine learning techniques were compared to make a comparison of the proposed model. Moreover, different test sets were created and the results were observed in a repeated manner. The results of these tests are given in detail in "Comparison of the performance of Deep Learning with other Machine Learning Algorithms" according to the performance measurement methods given below.

The true positive ratio (1), true negative ratio (2), false positive ratio (3), false negative ratio (4), and precision value (5), which are referred to as recall, are calculated as follows:

$$TPR = \frac{TP}{TP + FN} \tag{1}$$

$$TNR = \frac{TN}{TN + FP} \tag{2}$$

$$FPR = \frac{FP}{FP + TN} \tag{3}$$

$$FNR = \frac{FN}{FN + TP} \tag{4}$$

$$P = \frac{TP}{TP + FP} \tag{5}$$

TP stands for true positive sample amount, TN stands for true negative sample amount, FP stands for false positive sample amount, and FN stands for false negative sample amount. P, the precision value, stands for the ratio of malicious applications classified as true.

The F-measure value is measured according to Eq. (6) separately for both the malicious and benign datasets. These two calculations are made according to the weighted FM Eq. (7).

$$F - measure(FM) = \frac{2 \times recall \times precision}{recall + precision} \tag{6}$$

$$WFM = \frac{(F_{benign} \times N_{benign}) + (F_{malware} \times N_{malware})}{N_{benign} + N_{malware}} \tag{7}$$

## Deep learning classifier results

Performance measurement results according to the number of different hidden layers in the deep learning model are given in Table 2. The results were obtained using an input vector containing 349 input parameters. In seven different test models 2-, 3-, 4-, and 5-layered deep learning models were used. Thus, it was aimed to create the best performing deep learning model. Accordingly, although the results were very close to each other, the best results were achieved with a four-layer model containing 300, 300, 300 neurons. An average of 1 min 49 s was required for this training. In total 286,202 parameters were generated. The results were obtained from data divided into 80% training and 20% test sets. Training was provided over 50 epochs.

Both recall and precision values were above a certain level, so this indicated that the model was not good in one-way detection (only malicious detection or only benign detection) but it was successful in both cases. In classification problems, when the numbers of cluster elements are not evenly distributed, simply measuring the accuracy of the model is often an inadequate metric. For this reason, the performance of the proposed model was analyzed with precision, recall and F-measure values. The precision value was 99.24%. In malware detection, detecting benign practices as malicious can cause serious problems. For this reason, a high precision value shows that the model is successful in FP marking. Furthermore, a more successful value was obtained for the recall value, 98.78%. This also shows that it gives good results in detecting malware. The value of the F-measure at which precision and recall values are evaluated together and unbalanced cluster distributions can be observed was 98.9%. Successful results were obtained in this measurement in which all costs are evaluated. On the other hand, quite successful results

**Table 2 Deep learning results with different combination of hidden layers.**

| Total trainable parameters (DNN Model) | TPR (sensitivity) | TNR (specifity) | FPR | FNR | Precision | Recall | Accuracy | AUC | WFM | Runtime (min:sec) |
|---|---|---|---|---|---|---|---|---|---|---|
| 20152 (50,50) | 0.946 | 0.987 | 0.129 | 0.053 | 0.987 | 0.992 | 0.980 | 0.955 | 0.982 | 00:40 |
| 195902 (300,300) | 0.942 | 0.987 | 0.129 | 0.057 | 0.987 | 0.991 | 0.980 | 0.955 | 0.982 | 01:45 |
| 45352 (100,50,100) | 0.954 | 0.981 | 0.181 | 0.045 | 0.982 | 0.993 | 0.980 | 0.940 | 0.981 | 00:50 |
| 166002 (300,100,300) | 0.937 | 0.986 | 0.137 | 0.062 | 0.986 | 0.990 | 0.980 | 0.952 | 0.982 | 01:49 |
| 65502 (100,100,100,100) | 0.942 | 0.986 | 0.137 | 0.057 | 0.988 | 0.991 | 0.980 | 0.957 | 0.982 | 00:52 |
| 376502 (300,300,300,300) | 0.956 | 0.987 | 0.129 | 0.043 | 0.987 | 0.993 | 0.980 | 0.956 | 0.983 | 02:05 |
| 75602 (100,100,100,100,100) | 0.942 | 0.986 | 0.013 | 0.058 | 0.986 | 0.991 | 0.980 | 0.953 | 0.982 | 00:54 |

were produced even in the two-layer neural network with 50 neurons where the model is much simpler. For this reason, modeling can be performed using a simpler neural network depending on the intended use. However, in the present study, details are shown for the model at which the highest values were obtained.

When the tests were performed with 70% training and 30% test sets, precision, recall, accuracy, and F-measure values were 0.979, 0.992, 0.980 and 0.986, respectively. According to the tests performed with 80% training and 20% test sets, there was a 1% decrease in the results for some measurement metrics. This shows that the increase in the number of applications assigned for the training set will cause more successful results in the classification of tested applications.

The scheme of the model with the best results is shown in Fig. 3. According to this scheme, 349 parameters obtained from the features of mobile applications are given as inputs and a binary result is produced as a result of one input layer, three hidden layers, and one output layer. The complexity of the model is at a normal level and model quickly completes the training process. The input vector with 349 features is reduced to 300 in the first layer and training is carried out with 300 neurons up to the output layer. In the output layer, it is reduced to two for a dual classification. The model includes 286,202 trainable parameters. ReLu was used as the activation function in the hidden layers. Softmax was used as the activation function in the output layer. The optimizer that used the error back propagation phase was Adam. Increasing the complexity of the model slowed down the learning process; however, this did not provide a noticeable increase in classification performance. For this reason, a model that contained more hidden layers with more neurons was not designed. In addition, considering that this model will work on devices with limited resources such as mobile devices, it was thought to be beneficial to work with simpler models.

Figure 4 shows the change in accuracy on the training and test data over 50 epochs. As can be seen from the graphic, the proposed method overcame the over-fitting problem. Approximately after the first 10 epochs, it was seen that the model actually reached a certain level. However, a stable result was not produced in the training and testing phases. For this reason, the number of epochs was gradually increased and the aim was to obtain a more stable structure for the results.

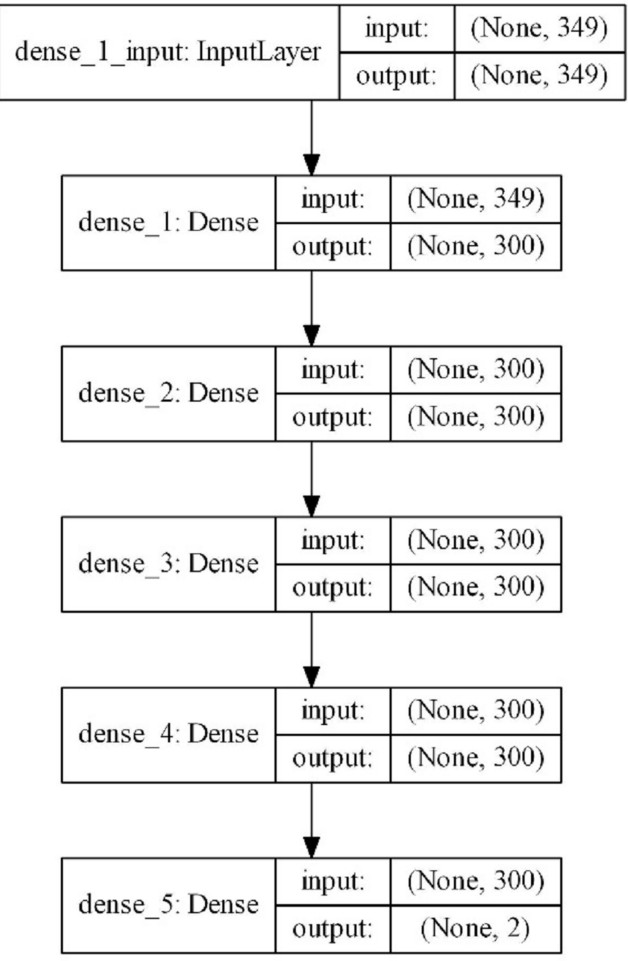

**Figure 3  Proposed DNN model layer structure.**   

The ROC curve is shown in Fig. 5 to observe the best results in consent-based classification. The ROC curve is used to measure the malware detection rate. It shows the effect of the learning model on the malware detection rate change and on the increase or decrease in the false positive value. The curve shows the change between the TP value and FP value and an increase in one value causes a decrease in the other. The fact that the ROC curve is close to the left and upper part, as shown in the figure, shows that the model gives the best results. The area under the curve is measured and gives the value of the deviation. Accordingly, a value of 0.9 and above is generally stated as a perfect classification (*Feizollah et al., 2017*) and it was 0.9515 in the present study. This result shows that the model was very successful in terms of classification in malware detection.

The confusion matrix of the test is shown in Fig. 6. High classification success was achieved in TP and TN values. However, the high values of the classification numbers for FP and FN indicates a very dangerous situation for end users. It will cause users not to use some useful and safe applications for no reason and, even worse, it will cause users to be at risk because some malicious applications are considered safe by them. FP and FN

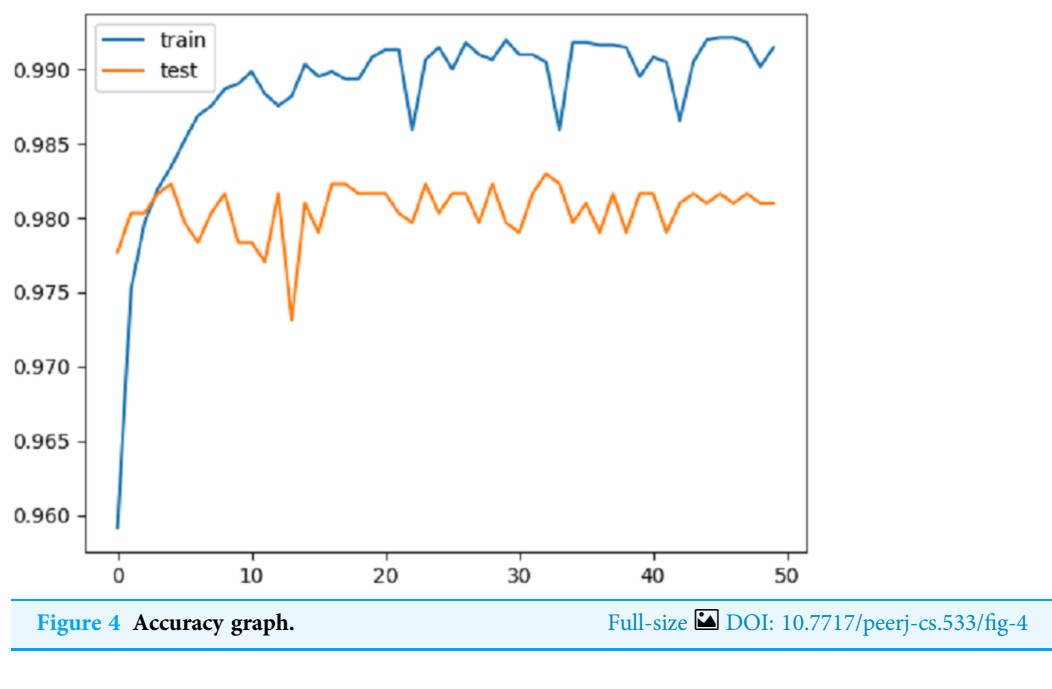

**Figure 4  Accuracy graph.**      

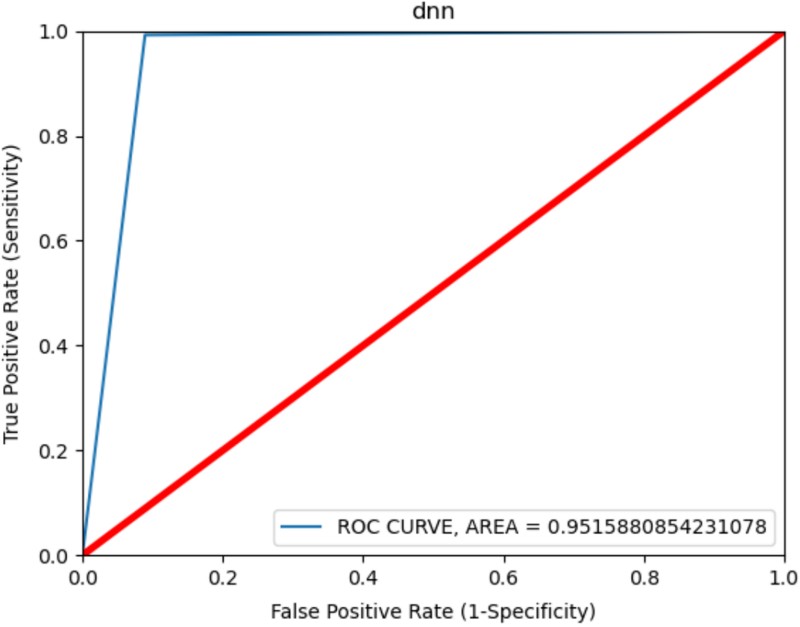

**Figure 5  ROC curve for Android permissions.**

were obtained in only 29 of the total 1,525 tests performed in the present study, demonstrating the success of this model.

## Comparison of the performance of deep learning with other machine learning algorithms

In this section, the accuracy of the proposed deep learning model and the results of traditional machine learning algorithms were compared. Nine different classification

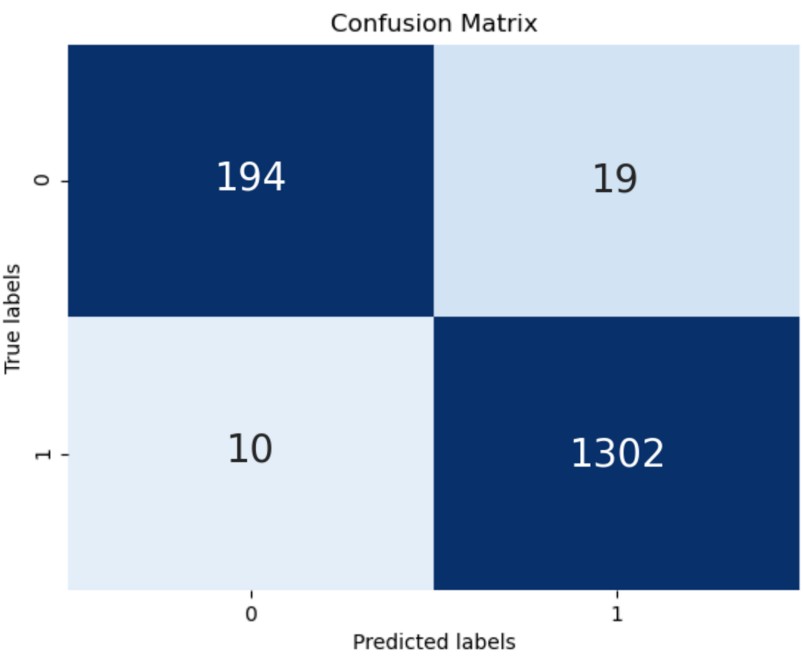

**Figure 6 Confusion matrix.**                                 

**Table 3 Results for 11 machine learning algorithms and deep learning.**

| Algorithms | TPR | TNR | FPR | FNR | Precision | Recall | Accuracy | WFM |
|---|---|---|---|---|---|---|---|---|
| KNeighbours | 0.8901 | 0.9777 | 0.022 | 0.109 | 0.8438 | 0.8901 | 0.9672 | 0.8663 |
| RF | 0.9489 | 0.9815 | 0.018 | 0.051 | 0.8698 | 0.9489 | 0.9777 | 0.9076 |
| SVC | 0.9588 | 0.9786 | 0.021 | 0.041 | 0.8490 | 0.9588 | 0.9764 | 0.9006 |
| Decision tree | 0.9535 | 0.9793 | 0.020 | 0.046 | 0.8542 | 0.9535 | 0.9764 | 0.9011 |
| GaussianNB | 0.7835 | 0.9188 | 0.081 | 0.216 | 0.3958 | 0.7835 | 0.9102 | 0.5260 |
| LinearDiscriminant | 0.9416 | 0.9657 | 0.034 | 0.584 | 0.7552 | 0.9416 | 0.9633 | 0.8382 |
| AdaBoost | 0.9419 | 0.9778 | 0.022 | 0.058 | 0.8434 | 0.9419 | 0.9738 | 0.8901 |
| GradientBoosting | 0.9689 | 0.9736 | 0.026 | 0.031 | 0.8125 | 0.9689 | 0.9731 | 0.8839 |
| ExtraTree | 0.9503 | 0.9851 | 0.014 | 0.104 | 0.8958 | 0.9503 | 0.9810 | 0.9223 |
| XGBoost | 0.9530 | 0.9800 | 0.020 | 0.046 | 0.8594 | 0.9530 | 0.9770 | 0.9041 |
| DL(376502(300,300,300,300)) | 0.9910 | 0.9870 | 0.029 | 0.043 | 0.9890 | 0.9910 | 0.9803 | 0.9820 |

algorithms were selected after several pre-tests and examinations and these are among the widely used techniques. Accordingly, it is shown in Table 3 that the deep learning model gave better results than the other classification models. Although the results were close to each other in general, overall results in the deep learning model were better. In the other classification algorithms, the weighted F-measure value was 0.9223 at most, while it was 98.90% in the deep learning model. Apart from the deep learning model, the most successful classification algorithms were ExtraTree, Random Forest (RF), and SVM. The successful results achieved with both machine learning algorithms and deep learning models showed that application features obtained with static analysis could produce quite favorable results in detecting malware.

**Table 4 The comparison of classification performance among former methods and proposed method.**

| Similar works | Selected features | Num of benign apps | Num of malware apps | Num of neurons or classification method | Precision | Recall | Accuracy | F-measure |
|---|---|---|---|---|---|---|---|---|
| ASAEF (*Zhang, Thing & Cheng, 2019*) | Metadata, permissions, intent filter, activity, services | 37,224 | 33,259 | N-gram, signature | 96.4% | 96.1% | 97.2% | 96.2% |
| FingerPrinting (*Karbab, Debbabi & Mouheb, 2016*) | Family DNA | 100 | 928 | Signature | 89% | 84% | N/A | 85% |
| DroidChain (*Wang et al., 2016*) | Permissions, API call, behaviour chain | – | 1,260 | Warshall | 91% | 92% | 93% | N/A |
| Shhadat (*Shhadat et al., 2020*) | Heuristic strategy, dynamic analysis | 172 | 984 | RF | 96.4% | 87.3% | 97.8% | 91.2% |
| DroidDet (*Zhu et al., 2018*) | Permissions, system events, sensitive API and URL | 1,065 | 1,065 | SVM | 88.16% | 88.40% | 88.26% | N/A |
| DL-Droid (*Alzaylaee, Yerima & Sezer, 2020*) | Application attributes, actions, events, permissions | 11,505 | 19,620 | 300, 100, 300 | 94.08% | 97.78% | 94.95% | 95.89% |
| SRBM (*Liu et al., 2021*) | Static and dynamic feature | 39,931 | 40,923 | RBM | – | – | 0.804 | 84.3% |
| Lu (*Lu et al., 2021*) | API calls | 1,400 | 1,400 | Correntropy, CNN | 95.0% | 76.0% | 84.25% | 84.0% |
| ProDroid (*Sasidharan & Thomas, 2021*) | API calls | 500 | 1,500 | HMM | 93.0% | 95.0% | 94.5% | 93.9% |
| Proposed model DL (376502(300,300,300,300)) | Application permissions | 961 | 6,661 | 300, 300, 300, 300 | 98.9% | 99.1% | 98.03% | 99.0% |

## Discussion

The model proposed in the present study is compared with similar deep learning or machine learning techniques used in previous studies in Table 4. It was observed that artificial intelligence modeling was used in almost all studies in 2019 and 2020. One of the main distinguishing differences among these studies is the dataset used and the second one is the feature vectors obtained from the applications in this dataset. In some studies, only the static property obtained from the manifest.xml file is used, while in other studies, intent filters, activity services, and API calls are used. The richness and homogeneity of the dataset are another factor with a direct effect on the results. In addition, the use of different classification methods was another reason for the difference in the results.

When the results were evaluated comparatively according to similar parameters, it was seen that the model proposed in the present study produced successful results with respect to other studies. While much better results were produced in some studies, better results were obtained with slight differences with the studies using similar modeling. Very good performance results were obtained according to studies with similar dataset sizes. This shows that existing classification performance values were taken one step further with the model proposed in the present study.

## CONCLUSIONS

The Android platform is the target of malicious mobile application developers and black hat hackers. Many antimalware tools aim to combat these applications and protect users.

In the present study, a model for Android malware detection was proposed. Models with high classification accuracy are needed in the development of this model. In these structures, there are two stages: selection of the features that best represent the problem and classification with high accuracy. In the current study, a deep neural network structure with three hidden layers to classify the permissions they request to represent applications was proposed. Reverse engineering applications were used to obtain feature vectors and a vector containing 349 features was obtained. The permissions requested by the applications are one of the most important parameters that reveal their purpose. The features used in the present study were obtained by static analysis having the advantages of low cost, high efficiency, and low risk. A total of 6,661 malicious samples taken from the Drebin and Genome datasets and 961 original benign application samples were used. In the experimental results, a 0.9924 precision value, 0.9878 recall value, 0.9816 accuracy and 0.9890 F-measure value were obtained. The results showed that Android permissions contain very good information for understanding the purposes of applications. The high classification performance obtained with the obtained dataset showed that the deep learning structure and Android permissions were a suggestion that could be adopted in the development of malware detection tools.

The model proposed in the present study can be developed easily and better performance results can be obtained with feature vectors using properties obtained by dynamic analysis. This will be part of my future work.

### Funding
The author received no funding for this work.

### Competing Interests
The author declares that he has no competing interests.

### Author Contributions
- Recep Sinan Arslan conceived and designed the experiments, performed the experiments, analyzed the data, performed the computation work, prepared figures and/or tables, authored or reviewed drafts of the paper, and approved the final draft.

### Data Availability
   The dataset is available as a Supplemental File.

### Supplemental Information
Supplemental information for this article can be found online at http://dx.doi.org/10.7717/peerj-cs.533#supplemental-information.

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
