# Peer review of "AndroAnalyzer: android malicious software detection based on deep learning"

_PeerJ Computer Science, doi:10.7717/peerj-cs.533_

## Round 0.1 · original submission · Major Revisions

· Academic Editor

Major Revisions

The paper needs improvements in terms of experimental results. The authors should state clearly the experiment parameters. Also, the comments of reviewers should be considered.

Reviewer 1 ·

Basic reporting

Although the idea is interesting, I recommend to improvement article with some comments as below:
- More description for experiments should be done.
- The method can be explained in a clearer way.Try to explain the theory more detailed in discussion.

Experimental design

- The reason of using certain parameters in the experiments should be discussed in more details.

Validity of the findings

no comment

Additional comments

- The references in this manuscript are somewhat out-of-date.In clude more recent researches in this field.
- The manuscript has not been carefully written.
- There are many grammar mistakes as well.
- The reason of using certain parameters in the experiments should be discussed in more details.

Reviewer 2 ·

Basic reporting

A clear view of the contribution is given in the paper.

Experimental design

The sample size differs from that used in related works. The authors, for instance, used 6661 Malware App, while the others used 33259, 928,... , and 19620. The distinction between these samples needs to be illustrated, so we can ensure that the authors have not compared the accuracy of basic malware apps with sophisticated ones used in similar works.

Validity of the findings

The accuracy values written for the work in Table 4 are in the wrong format. (e.g., 0.989 rather than %98.9).
Also, the header for this row is confused "Proposed Model DL(376502(300,300,300,300))"

Additional comments

None.

Reviewer 3 ·

Basic reporting

The paper describes a model using AndroAnalyzer that uses static analysis and a deep learning system is proposed. The model is tested in numerous applications, and additional tests were conducted with machine learning. The authors claim that 98.16% accuracy value was reached compared to 26 traditional machine learning techniques, where Precision and F-measure were 98.78, 27 99.24, and 98.90, respectively.
The paper is generally good. The motivation and research problem are well defined. The methodology is clear and the results are discussed and compared with some similar tools.
The paper needs more linguistic and grammatical improvement. It should proofread.
In the paper structure at the end of the Introduction Section, preplace the word “Chapter” with the word “Section”.
Heading and Sub-Heading are not numbered.
The Literature Review should be a standalone section named related.
Many papers are surveyed and classified, however, the studies reviewed are in between 2018-2020, why no mention of the studies contacted before 2018.
Change the sentence “In the [13] numbered study performed by Shahriar et al., a model was proposed to detect malicious software ….” to be “In [13], a model was proposed to detect malicious software …. ” and use this throughout the paper.
The description of the existing studies is not enough to shed light on the research gap. The existing work has to be compared and critically evaluated.
The methodology needs more clarification about its stages, and the figure should be linked with the stages described in detail in numbered subheadings and bearing the numbers of the stages in the illustrated figure. As well as clarify what is meant by some of the acronyms and not leave an indulgence for the reader to devise them

Experimental design

The experiments are conducted, explained, discussed, and evaluated with other related approaches

Validity of the findings

Results are explained, discussed, and evaluated with other related approaches

---

## Round 0.2 · Minor Revisions

· Academic Editor

Minor Revisions

As mentioned in the first review, the paper needs proofreading since it has grammatical errors.

Reviewer 2 ·

Basic reporting

None

Experimental design

None

Validity of the findings

None

Additional comments

The contribution is clear and significant, but still the Language needs to be improved. For example, the following statement in the result section “ Both obtaining the best deep learning structure ” is confused.

---

## Round 0.3 · accepted · Accept

· Academic Editor

Accept

It's clear that the revised paper is enhanced and so it's recommend to accept the paper

Reviewer 3 ·

Basic reporting

The paper describes a model using AndroAnalyzer that uses static analysis and a deep learning system is proposed. The model is tested in numerous applications, and additional tests were conducted with machine learning. The authors claim that 98.16% accuracy value was reached compared to 26 traditional machine learning techniques, where Precision and F-measure were 98.78, 27 99.24, and 98.90, respectively.
The paper is generally good. The motivation and research problem are well defined. The methodology is clear and the results are discussed and compared with some similar tools.
The paper needs more linguistic and grammatical improvement. It should proofread.
In the paper structure at the end of the Introduction Section, preplace the word “Chapter” with the word “Section”.
Heading and Sub-Heading are not numbered.
The Literature Review should be a standalone section named related.
Many papers are surveyed and classified, however, the studies reviewed are in between 2018-2020, why no mention of the studies contacted before 2018.
Change the sentence “In the [13] numbered study performed by Shahriar et al., a model was proposed to detect malicious software ….” to be “In [13], a model was proposed to detect malicious software …. ” and use this throughout the paper.
The description of the existing studies is not enough to shed light on the research gap. The existing work has to be compared and critically evaluated.
The methodology needs more clarification about its stages, and the figure should be linked with the stages described in detail in numbered subheadings and bearing the numbers of the stages in the illustrated figure. As well as clarify what is meant by some of the acronyms and not leave an indulgence for the reader to devise them
Experiments Results are explained, discussed, and evaluated with other related approaches

Experimental design

Experiments Results are explained, discussed, and evaluated with other related approaches

Validity of the findings

The existing work has to be compared and critically evaluated.
Results are explained, discussed, and evaluated with other related approaches

Additional comments

The Paper seems good but needs some modification